# In Vitro Hepatic Assessment of Cineole and Its Derivatives in Common Brushtail Possums (*Trichosurus vulpecula*) and Rodents

**DOI:** 10.3390/biology10121326

**Published:** 2021-12-15

**Authors:** Ravneel R. Chand, Mhairi Nimick, Belinda Cridge, Rhonda J. Rosengren

**Affiliations:** 1Department of Pharmacology and Toxicology, School of Biomedical Sciences, University of Otago, Dunedin 9016, New Zealand; mhairi.nimick@otago.ac.nz; 2Science for Communities, Christchurch Science Centre, Christchurch 8041, New Zealand; belinda.cridge@esr.cri.nz

**Keywords:** cineole and derivatives, brushtail possum, rodents, CYP3A, UGT

## Abstract

**Simple Summary:**

Folivore marsupials can metabolise high levels of dietary terpenes compared to eutherian mammals, indicating that they possess highly efficient drug metabolising enzymes. However, two such highly efficient drug metabolising enzymes, cytochrome P450 3A and UDP-glucuronosyltransferase, are involved in terpene metabolism, evidence for inhibitory actions by these dietary terpenes on these enzymes are limited. Therefore, we investigated the effect of cineole, a major component of eucalyptus essential oils, and its derivatives on these hepatic drug metabolising enzymes in mice, rats, and possums. Our findings revealed no major inhibition on these drug metabolising enzymes by cineole and its derivatives at concentrations up to 50 µM. However, species-specific difference in their basal enzymatic activity was observed and may provide an avenue for developing future pest control strategies.

**Abstract:**

Folivore marsupials, such as brushtail possum (*Trichosurus Vulpecula*) and koala (*Phascolarctos cinereus*), can metabolise higher levels of dietary terpenes, such as cineole, that are toxic to eutherian mammals. While the highly efficient drug metabolising enzymes, cytochrome P450 3A (CYP3A) and phase II conjugating enzymes (UDP-glucuronosyltransferase, UGT), are involved in the metabolism of high levels of dietary terpenes, evidence for inhibitory actions on these enzymes by these terpenes is scant. Thus, this study investigated the effect of cineole and its derivatives on catalytic activities of hepatic CYP3A and UGT in mice, rats, and possums. Results showed that cineole (up to 50 µM) and its derivatives (up to 25 µM) did not significantly inhibit CYP3A and UGT activities in mice, rats, and possums (both in silico and in vitro). Interestingly, basal hepatic CYP3A catalytic activity in the possums was ~20% lower than that in rats and mice. In contrast, possums had ~2-fold higher UGT catalytic activity when compared to mice and rats. Thus, these basal enzymatic differences may be further exploited in future pest management strategies.

## 1. Introduction

Eucalypt trees (*Eucalyptus*, *Corymbia*, and *Angophora*), widespread in the Australian landscape, are an easy food resource for marsupials such as brushtail possums (*Trichosurus vulpecula*) and koalas (*Phascolarctos cinereus*) [1]. These marsupials survive on eucalypt foliage despite containing abundant terpenes that would otherwise cause toxicity to other animals upon ingestion. For example, the intrinsic clearance of *p*-cymene (monoterpene found in Eucalyptus spp.) in terpene-treated possums was threefold higher than in rats, indicating a greater capacity to metabolise dietary terpenes [2]. Cineole (eucalyptol, 1,3,3-trimethyl-2-oxabicyclo [2.2.2]octane, 1,8-cineole), a main component of eucalyptus essential oils (concentration up to 84.4%) [3], was toxic following ingestion of lower doses in humans (<1 g kg^−1^) [4], dogs (1.5 g kg^−1^), mice (50 mg kg^−1^) [5], and rats (2.5 g kg^−1^) [6], compared to possums (3.6 g kg^−1^) [7]. The high level of cineole tolerance in possums is likely due to their unique metabolism [8] and adaptation [9].

Possums rely on oxidation by cytochrome P450 3A (CYP3A) and by conjugation using uridine 5’-diphospho-glucuronosyltransferase (UDP-glucuronosyltransferase, UGT) for rapid metabolic clearance of eucalypt monoterpenes and other secondary metabolites [2,8,10]. The CYP3A subfamily receives a greater interest compared to other drug metabolising enzymes because of its abundant expression in the liver [11]. Previous studies have also showed that CYP3A enzymes metabolise cineole differently; where regioselectivity is favoured in different species. For instance, C9 carbon oxidation of cineole was favoured in possums over C3 and C2 oxidation in humans and rats [8].

Although CYP3A is involved in the metabolism of plant secondary metabolites, other studies have reported that possums depend to a lesser extent on CYP3A than other species. This was evident when the metabolism of midazolam, a specific substrate of CYP3A, in possum liver was lower than that reported in human liver [12]. Specifically, the *V*_max_ values (*p* mol/mg/min) of 4-hydroxymidazolam and 1′-hydroxylmiazolam in possum liver microsomes were 23- and 8-fold lower than those observed in human liver microsomes. However, very little information is available on the ability of possums to metabolise xenobiotics when compared with other species. Such investigations are important to understand the specificity of drug metabolising enzymes. This information could be used to improve the specificity of oral contraceptives for possum management. Therefore, to better understand the activities of CYP3A and UGT, we evaluated their catalytic activity and protein levels in possums and rodents (mice and rats). This is the first in vitro study to compare the effect of cineole and its derivatives on these enzymes in possums and rodents.

## 2. Materials and Methods

### 2.1. Chemicals

Nicotinamide adenine dinucleotide phosphate (NADP^+^), uridine 5′-diphosphoglucuronic acid trisodium salt (UDPGA), dimethyl sulfoxide (DMSO), bovine serum albumin (BSA), Eucalyptol or cineole (99%) and its derivatives including *trans*-terpin, 1,1-dimethylcyclohexane, dimethyl-*trans*-cyclohexane-1,4-dicarboxylate, 7-oxabicyclo[2.2.1]-hept-5-ene-2-carboxylic acid, *cis*-1,5-dimethylbicyclo[3.3.0]octane-3,7-dione, 4-pentylbicyclo[2.2.2]octane-1-carboxylic acid, and camphor were purchased from Sigma-Aldrich (St.Louis, MO, USA). The following primary antibodies were purchased; CYP3A polyclonal antibody (PA5-14896, Invitrogen, Waltham, MA, USA), UGT2B4 (PA5-92155, Invitrogen, Waltham, MA, USA), and GAPDH (Sigma-Aldrich, Sydney, Australia; an affiliate of Merck KGaA, Darmstadt, Germany). All other chemicals/reagents used were of highest purity commercially available.

### 2.2. Selection of Cineole Derivatives

Derivatives of cineole were screened for their effect on CYP3A and UGT catalytic activities. This screen involved database searches across DrugBank, PubChem, ChemSpider, and ADMET Spider (a toxicological predictive database) for core structural similarities. Searches particularly focused on the bicyclooctane functional group in cineole’s core molecule. This was widened to include cyclohexane functional groups, with or without the carboxylic functional group (Figure 1). Only market approved drugs were included in the search criteria. From this, target compounds having structural features likely to result in inhibition of CYP3A in possums were identified. Compounds with mild toxicity or expected toxicity at occupational levels were included but compounds with environmental or significant acute toxicity in mammals were excluded.

### 2.3. Animals

Five to eleven possums (*Trichosurus vulpecula*) were collected by staff at Manaaki Whenua—Landcare Research from Okuti Valley, Banks Peninsula (Christchurch, New Zealand). Possum trapping and liver harvesting was approved by the Manaaki Whenua—Landcare Research Animal Ethics Committee. Possums were leg-hold trapped, humanely killed, and livers were harvested, snap frozen in liquid nitrogen, and stored in a −80 °C freezer until the preparation of microsomal fractions. Four Male Balb/c mice (6–8 weeks old) and 5 male Sprague Dawley rats (2–3 months old), were purchased from the Hercus Taieri Resource Unit (Dunedin, New Zealand). Mice and rats were euthanised by carbon dioxide inhalation and livers were removed and microsomes were prepared. Mouse and rat experiments were approved by the University of Otago Animal Ethics Committee (AEC Approval Numbers 20/26 and 19/10). Possum tissues were gifted from Landcare Research, Christchurch, New Zealand.

### 2.4. Preparation of Mouse, Rat, and Possum Liver Microsomes

Microsomal preparation was as previously outlined [13]. Individual livers were placed in beakers containing 1.15% KCl. All the procedures were carried out at 0–4 °C. The liver samples were minced and homogenised in buffer A (0.1 M Tris, 0.1 M potassium chloride, 1 mM EDTA, and 20 μM butylated hydroxytoluene, at pH = 7.4) using the Teflon-glass homogeniser (5–6 vertical passes). The samples were centrifuged for 20 min at 10,000× *g*. The supernatant was then transferred into another set of centrifuge tubes and spun for 60 min at 100,000× *g*. The pellet was resuspended in Buffer B (0.1 M potassium pyrophosphate, 1 mM EDTA, and 20 μM butylated hydroxytoluene, at pH = 7.4), homogenised and recentrifuged for 60 min at 100,000× *g*. Finally, the pellet was homogenised in 1–2 mL of Buffer C (10 mM Tris-HCl, 1 mM EDTA, and 4% glycerol (*v*/*v*)) and stored at 80 °C until used. The protein concentration of the microsomes was determined using the Bicinchoninic acid (BCA) method [14].

### 2.5. CYP3A Catalytic Activity

CYP3A catalytic activity was determined using the erythromycin-*N*-demethylation activity assay as described previously [15]. The reaction mixture (final volume of 1 mL and final protein concentration of 1 mg) contained 500 µL of microsomes in 0.1 M phosphate buffer (pH = 7.4), 400 µL of erythromycin buffer (0.45 mM), and 50 µL of the test compound was pre-warmed at 37 °C for 2 min. The reaction was initiated with 50 µL of 2.5 mM NADP^+^ and incubated at 37 °C for 30 min (optimised incubation time). The reaction was stopped with the addition of 330 µL of 15% zinc sulphate. Tubes were vortexed and left for 5 min (at room temperature) and then 330 µL of saturated barium hydroxide was added. Tubes were further vortexed and left for 5 min (at room temperature). Tubes were then centrifuged at 1800 *g* for 10 min. To 830 µL of the supernatant, 330 µL of Nash reagent (30% (*w*/*v*) ammonium acetate, 0.4% (*v*/*v*) acetyl acetone) was added and further incubated at 60 °C for 30 min. The blank tube contained 330 µL of Nash reagent and 830 µL of Milli-Q-water. The tubes were centrifuged again at 1800× *g* for 10 min and the absorbance was read at 415 nm. Results presented are the mean ± SEM of 3 independent experiments conducted in duplicate.

### 2.6. Prediction of CYP Inhibition In Silico

CYP3A inhibition activities using cineole and its derivatives were further predicted with DL-CYP Prediction server [16]. This is a web-based tool that predicts the inhibition tendency of small molecules on five major CYP isoforms, namely CYP1A2, CYP2C9, CYP2C19, CYP2D6, and CYP3A4. The server uses a deep autoencoder multi-task neural network. All the chemical structures (sdf format) were downloaded from PubChem (National Centre for Information) and were used to predict the tendency to inhibit CYP3A4 enzyme (most abundantly expressed isoform in hepatic microsomes). The AC_50_ value (concentration that gives 50% activation of the enzyme) was used as one of the main criteria to group compounds either as an inhibitor or noninhibitor [17]. Compounds with AC_50_ value ≤10 µM were grouped as inhibitors while compounds with AC_50_ value >57 µM were considered noninhibitors. In addition, ketoconazole was used as a positive control and all results are expressed as values from 0 to 1.

### 2.7. Glucuronidation of p-Nitrophenol

The rate of *p*-nitrophenol glucuronidation was determined as previously reported [18]. In short, the incubation mixture (500 µL in total) contained; 100 µL of 1 mg/mL microsomes, 50 µL of 1 M Tris-HCl (pH = 7.4), 20 µL of 0.25% (*v*/*v*) Triton-X 100, 50 µL of 50 mM MgCl_2_, 130 µL of double distilled water, 50 µL of 5 mM *p*-nitrophenol, 50 µL of the test compound, and 50 µL of uridine 5′-diphosphoglucuronic acid trisodium salt (UDPGA). The blank tubes had all components except UDPGA and microsomes; instead, 150 µL of 0.1 M phosphate buffer was added to equal the volume as the reaction mixture. The reaction mixture and the blank were incubated at 37 °C for 2 min. Then, 50 µL of 30 mM UDPGA was added to initiate the reaction which proceeded at 37 °C for 30 min (optimised incubation time). The reaction was stopped using 1 mL of 5% trichloroacetic acid (TCA). The tubes were centrifuged at 1800 *g* for 10 min and then 1 mL of supernatant was added to 250 µL of 2 M sodium hydroxide (NaOH). The absorbance was read at 405 nm. The sample reading was subtracted from the blank absorbance and divided by the extinction coefficient of *p*-nitrophenol (18.1 × 10^3^ cm^2^/mol) and the light path of the spectrophotometer (1 cm). The glucuronidation activities are expressed in µmol/mg/min and are the mean ± SEM of 3 independent experiments conducted in duplicate.

### 2.8. Western Blotting

Microsomes were diluted to 1 mg/mL in 4× Laemmli buffer (62.5 mM Tris-HCl, 1% sodium dodecyl sulphate, 10% glycerol, 0.005% bromophenol blue, and 355 mM β-mercaptoethanol). The samples were then heated at 95 °C for 5 min and stored at −20 °C. Gel electrophoresis was carried out as described previously [19]. All gel casting and running was performed using a Mini PROTEAN^®^ (BioRad). Microsomal samples were heated at 37 °C before loading 1–2 µg of protein onto a 7.5% acrylamide gel (acrylamide, 1.5 M lower Tris, 50% glycerol, 10% ammonium persulphate, and tetramethylethylenediamine), and Precision Plus Protein Dual Colour Standards (Bio-Rad Laboratories, Hercules, CA, USA) was loaded as a protein ladder. Empty wells were loaded with 15 μL of 1× sample buffer. Gels were run at 80 V for protein stacking (~15 min) and then 120 V to resolve the protein (~1.5 h). Following the separation, proteins were transferred to a PVDF membrane (Merck Millipore Ltd., New Zealand) in transfer buffer at 100 V for 60 min. The membranes were blocked in 5% non-fat milk and TBST (Tris-buffered saline with 1% Triton-X 100) for 1 h. The membranes were then incubated with appropriate primary antibody in TBS (Tris-buffered saline) and left overnight on a shaker at 4°C. The following antibodies were used: CYP3A polyclonal antibody (1:1000, PA5-14896, Invitrogen, USA), UGT2B4 (1:1000, PA5-92155, Invitrogen, Waltham, MA, USA), and GAPDH (1:2000, Sigma-Aldrich, Sydney, Australia; an affiliate of Merck KGaA, Darmstadt, Germany). The membranes were then stripped and washed with TBST buffer (6 × 5 min). Horseradish peroxidase-conjugated secondary antibody (anti-rabbit for CYP3A and UGT2B7, and anti-mouse for GAPDH) in TBS and milk powder was added to the membranes and were incubated for 45 min at room temperature. The membranes were re-washed (6 × 5 min) with TBST buffer before the addition of the SuperSignal West Pico Chemiluminescent substrate or Femto (ThermoFisher, Albany, New Zealand), and blots were visualised using a CL-XPosure Film (ThermoFisher, Albany, New Zealand).

### 2.9. Statistical Analysis

Statistical significance was assessed using a one-way ANOVA with a Dunnett’s post hoc test, where *p* < 0.05 was the minimum requirement for a statistically significant difference. All the graphs and statistical analysis were performed using GraphPad Prism software version 9.0 for Windows (GraphPad Software, San Diego, CA, USA).

## 3. Results

### 3.1. CYP3A Catalytic Activity

First, we determined the basal CYP3A catalytic activities using mouse, rat and possum hepatic microsomes. The mean basal CYP3A catalytic activities were 0.69 ± 0.01, 0.59 ± 0.02, and 0.38 ± 0.004 nmol/mg/min, respectively. The results showed that mouse and rat had significantly higher CYP3A hepatic catalytic activities (21–28%) compared to possums (Figure 2A). However, CYP3A protein levels in possums were significantly higher (~2-fold) compared to mouse and rat (Figure 2B,C; also see Appendix A). Furthermore, the effect of cineole and eight other derivatives on CYP3A catalytic activity was then determined. No significant alteration/inhibition of CYP3A catalytic activities was observed in the three species at concentration of 12.5 µM and 25 µM (Table 1). These concentrations were estimated from a previous study on regulation of CYP450 enzymes using bioactive components of plant extracts [20]. Moreover, to ensure that the lack of inhibition was not related to concentration, the parent compound cineole was examined at a twofold higher concentration (50 µM) in all species. The results showed no major inhibition, as the activity in mice only decreased to 89.3% of control. However, when ketoconazole (5 µM) was used as a positive control the mouse hepatic catalytic activity decreased by 60 percent control (Figure 3).

### 3.2. Prediction of CYP Inhibition In Silico

The potential for CYP3A4 inhibition by cineole and/or its derivatives was predicted with the DL-CYP Prediction server. The results showed that cineole and its derivatives had no predicted inhibitory effects as the predicted value was 0 (Table 2). Importantly, ketoconazole (positive control) was predicted accurately with a value of one, indicating that it is a potent CYP3A4 inhibitor [21].

### 3.3. p-Nitrophenol Glucuronidation

We first examined the basal activity of UGT in the three test species to determine if the species-specific differences would be seen. The results showed that possums had ~2-fold higher hepatic *p*-nitrophenol glucuronidation activity compared to mice and rats (Figure 4A). However, protein levels of the UGT2B4 isoform in the mouse was >2-fold higher than that in rat and possum liver, as determined by Western blotting (Figure 4B,C; also see Appendix A). Furthermore, none of the compounds showed any significant inhibition towards *p*-nitrophenol glucuronidation. As shown in the Table 3 at the highest tested concentration (25 µM), all compounds had catalytic activity at 90% of control or higher. Further incubation with cineole at a higher concentration (50 µM) also failed to significantly inhibit *p*-nitrophenol glucuronidation (Figure 5). However, when fluconazole (2.5 mM) was used as a positive control the mouse hepatic *p*-nitrophenol glucuronidation decreased by 20 percent control.

## 4. Discussion

### 4.1. CYP3A Catalytic Activity

Basal hepatic CYP3A catalytic activities in mice and rats were higher compared to possums. However, analysis by Western blotting indicated that possums had ~2-fold higher CYP3A protein levels in hepatic microsomes compared to mice and rats; indicating that catalytic activity differs compared to the protein level in these species. This could have possibly caused by variation in protein phosphorylation; a mechanism regulating the protein activity [22]. Moreover, there is no direct evidence that supports the basal CYP3A catalytic activities in possums; however, Sorensen, et al. [23] reported that possums express at least three different CYP3A-like isoforms (CYP3A P1, P2, P3) in their liver and duodenum that may contribute to a high tolerance for a wide range of xenobiotics. In mice, the basal level of hepatic CYP3A catalytic activities was similar (<1 nmol/min per mg protein) to those reported previously [24,25]. Even though there were basal differences, none of the tested compounds up to 50 µM showed any significant inhibition of CYP3A catalytic activity in mouse, rat, and possum hepatic microsomes. However, greater cineole concentration (>1 mM) may potentially reduce the catalytic activity [26]. Moreover, in silico study using the DL-CYP Prediction Server has further supported that cineole and derivatives have no inhibitory action on the CYP3A4 isoform. The predicted values for cineole and its derivatives on CYP3A4 inhibition were zero; indicating that these compounds would reduce catalytic activity at higher concentrations (>57 µM) and therefore were considered to be noninhibitors.

Several studies have suggested that CYP3A isoforms are the likely enzyme responsible for terpene metabolism in possums [23,27]. Specifically, Pass, et al. [8] revealed that possums heavily rely on CYP enzymes (preferably CYP3A) for the oxidation of cineole. The conversion of cineole to 9-hydroxycineole accounted for 75% of the total intrinsic clearance in possums, while 3-hydroxycineole and 7-hydroxycineole accounted for 14% and 11%, respectively. Further studies have also assessed the inhibition of cineole metabolism in possums using specific inhibitors of rat and human CYPs. Ketoconazole (a competitive and specific CYP3A inhibitor) reduced the metabolism of cineole in both possum and rat microsomes [28]. They reported that ketoconazole (5 µM) reduced the formation of 9-hydroxycineole in possums by 15% at lower cineole concentration (10 µM) and 30% at a higher cineole concentration (100 µM); however, in rats, ketoconazole reduced the metabolite formation by 85%. Similarly, Miyazawa, et al. [29] also suggested that cineole is one of the effective substrates for CYP3A enzymes in the rat and human liver microsomes. Thus, showing that CYP3A enzymes are actively engaged in the metabolism of cineole in various species.

Although there is a widespread acknowledgement that CYP3A is involved in terpene metabolism, disagreement remains on the CYP3A induction by terpene. Data from Sorensen, et al. [23] suggested that CYP3A-like enzymes are not induced by terpenes. However, it was clear from extensive evidence in the literature that other CYPs (such as the CYP2 and CYP4 families) are induced by dietary terpenes. Pass, et al. [10] showed that terpenes (such as cineole, *p*-cymene, α-pinene, and limonene) caused overall induction in CYPs (50% higher total CYP450 content) and a 45% increase in aminopyrine demethylase activities in possums. This was further confirmed by Western blotting using human CYP2E1, rat CYP2C11 and CYP2C6 antibodies where greater immunoreactive bands were seen in treated possums. A further study by Ngo, et al. [30] also suggest that hepatic CYP4A is induced in possums, where the peroxisomal cyanide-intensive palmitoyl coenzyme A oxidative activity was significantly higher (>2-fold) in possums on a terpene diet compared to the possum group fed a fruit and cereal diet. Overall, CYP3A isoforms are involved in terpene metabolism; however, whether all CYPs (particularly CYP3A) are induced or not by terpenes remain questionable. However, we have clearly shown that CYP3A is not inhibited by cineole and its derivatives.

Further research is needed to verify exactly how CYP3As interact with dietary terpenes. Several factors may influence the study, particularly the validity of samples used (notably possum samples). In most of the in vivo studies researchers have utilised wild possums that were captured and studied up to 30 days later [10,23]. However, captive-bred animals should be used for toxicity studies to reduce any factors from the environment, genetic and other factors such as disease and age [31,32].

### 4.2. p-Nitrophenol Glucuronidation

The UGT enzymes catalyse covalent linkage of glucuronic acids to its metabolite (via conjugation), making the metabolites more hydrophilic for easier excretion [1]. The present study showed that the basal *p*-nitrophenol glucuronidation activities in the possum were ~2-fold higher than those in mouse and rat hepatic microsomes; indicating a greater capacity to metabolise phenolic compounds. Higher *p*-nitrophenol glucuronidation in possum liver microsomes may also explain why marsupial folivores such as the koala can rapidly excrete 2–3 g of glucuronic acids daily [33]. Moreover, the basal *p*-nitrophenol glucuronidation activities were comparable to the literature data on mouse and rat hepatic microsomes [13,34]. Similar to CYP3A activity no significant effect of cineole and/or its derivatives was seen on *p*-nitrophenol glucuronidation. This confirms that cineole and its derivatives did not inhibit the glucuronidation of *p*-nitrophenol at concentrations up to 25 µM.

*P*-nitrophenol is a well-known phenolic substrate used for measuring glucuronidation activities [35] despite other studies suggesting overlapping substrate specificity within the UGT family. Furthermore, species difference in basal UGT activity has been reported [36,37]. This was also indicated in the present study, where mice had ~2-fold higher UGT2B4 expression compared to rats and possums. Specifically, *p*-nitrophenol is known to be metabolised by other UGT isoforms such as UGT1A1, UGT1A3, UGT1A4, UGT1A6, UGT1A9, UGT2B7, and UGT2B17. This is not a new phenomenon as overlapping activity of a range of other substrates on UGT isoforms was reviewed by Tukey and Strassburg [38]. Since this was a comparative study between possums and rodents, using *p*-nitrophenol as a substrate for UDP-glucuronosyltransferases was considered useful as a rapid and an easy screen. Further investigation with specific isoforms of phase II enzymes involved in terpene metabolism may be warranted to further understand the adaptation of possums (and other marsupials) towards this chemical class. Furthermore, glucuronic acid has been detected in possum’s urine following digestion of dietary terpenes [39]. It was observed that cineole and terpineol in the diet increased the excretion of glucuronic acid to 0.1–1–5 mmol/day. Glucuronidation is also an important pathway for excretion of phenolic plant secondary metabolites in koalas, where koalas can excrete up to 2–3 g of glucuronic acid daily and these are mainly conjugated with phenols and aglycones [33]. Other studies have also shown varying glucuronidation activities between species, such as dogs, which lack N-acetylation capacities [40], and cats, which have no glucuronidation capacities [41]. Much of the work in this area is still limited and warrants further exploration of xenobiotic metabolism in marsupial vs eutherian mammals.

## 5. Conclusions

This is one of the first in vitro studies directly comparing the effect of cineole and its derivatives on hepatic CYP3A and *p*-nitrophenol catalytic activity in mice, rats, and possums. Our findings revealed that cineole (up to 50 µM) and its derivatives are not potent inhibitors of CYP3A or UGT. In silico modelling further supported that cineole and its derivatives have no inhibitory effects on CYP3A catalytic activities. Long-term drug administration studies would be required to determine if any of these compounds could induce the drug metabolising enzymes. Collectively, mice and rats were found to have higher basal CYP3A catalytic activities compared to possums. However, possums had a higher basal glucuronidation of *p*-nitrophenol compared to mice and rats; indicating a greater capacity to metabolise phenolic compounds. These basal enzymatic differences may provide an avenue for developing future pest control strategies.

## Figures and Tables

**Figure 1 biology-10-01326-f001:**
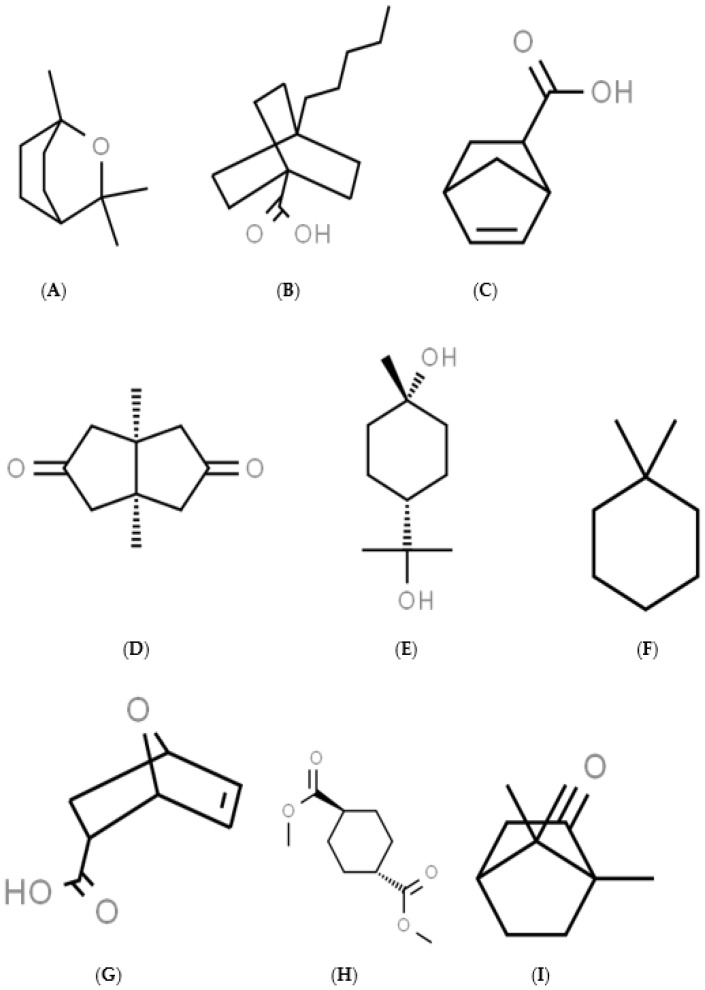
**The structures of the 9 chemicals studied**: Cineole (**A**), 4-pentylbicyclo[2.2.2]octane-1-carboxylic acid (**B**), 5-norbornene-2-carboxylic acid (**C**), *cis*-1,5-dimethylbicyclo[3.3.0]octane-3,7-dione (**D**), *trans*-terpin (**E**), 1,1-dimethylcyclohexane (**F**), 7-oxabicyclo[2.2.1]-hept-5-ene-2-carboxylic acid (**G**), dimethyl-*trans*-cyclohexane-1,4-dicarboxylate (**H**), and camphor (**I**).

**Figure 2 biology-10-01326-f002:**
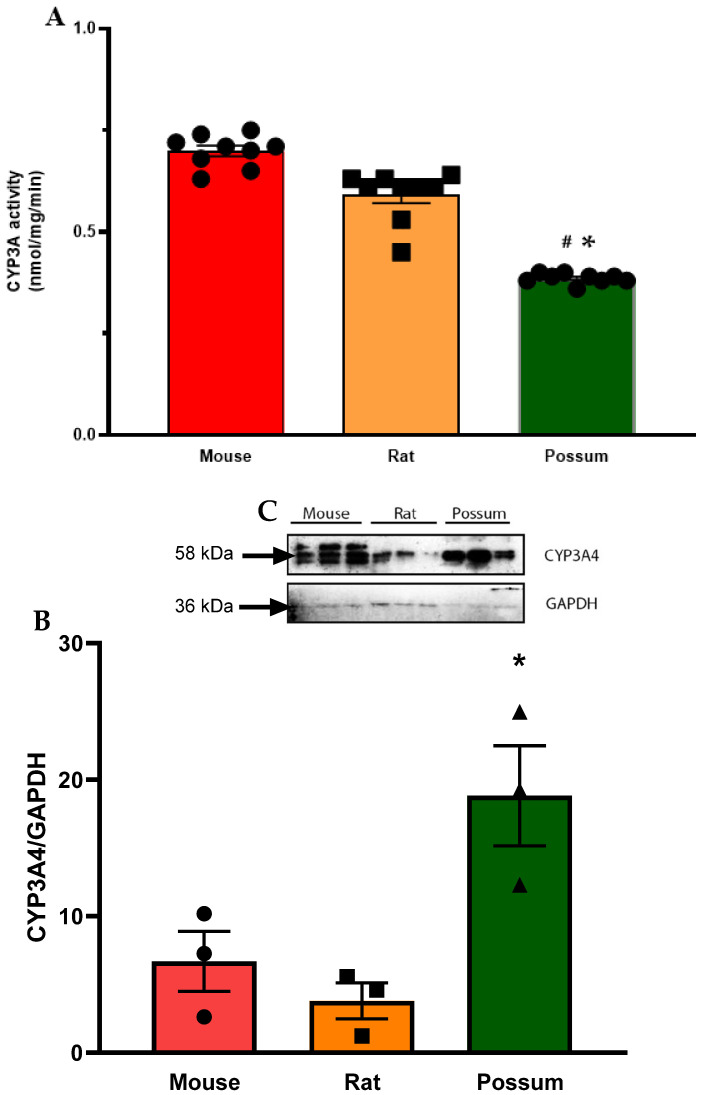
**Hepatic CYP3A basal activity and protein levels in mouse, rat, and possum**: (**A**) Hepatic microsomes from each species were incubated with 0.1 M phosphate buffer and <0.5% DMSO, and erythromycin-N-demethylation was determined as an indicator of CYP3A catalytic activity. Values are expressed as nmol/mg/min. Bars represent the mean ± SEM from basal catalytic activities in each species. (**B**) Protein content for CYP3A in hepatic microsome from each species was determined by Western blotting (loaded protein in each well was 2 µg). Results from scanning densitometry of CYP3A Western blot normalised to the housekeeper protein GAPDH. The bars represent the mean ± SEM of the optical density from n = 3. (**C**) Representative image for CYP3A Western blot from left to right: mouse, rat and possum. * and # significantly different compared to solvent control in the mouse and rat, *p* < 0.05, respectively.

**Figure 3 biology-10-01326-f003:**
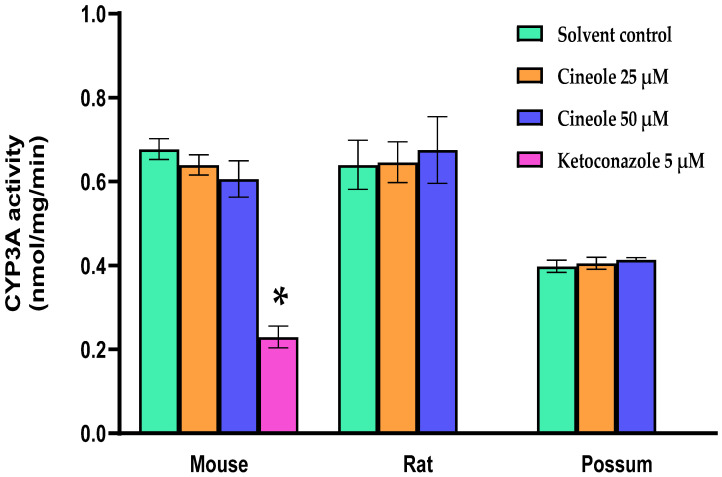
**CYP3A catalytic activity of cineole in mouse, rat, and possum**. Hepatic microsomes from each species were incubated with cineole (25 and 50 µM) and erythromycin-*N*-demethylation was determined as an indicator of CYP3A catalytic activity. Values are expressed as nmol/mg/min. Ketoconazole was used as a positive control. Bars represent the mean ± SEM from n = 3 in duplicate. * Significantly decreased compared to solvent control in the mouse, *p* < 0.05.

**Figure 4 biology-10-01326-f004:**
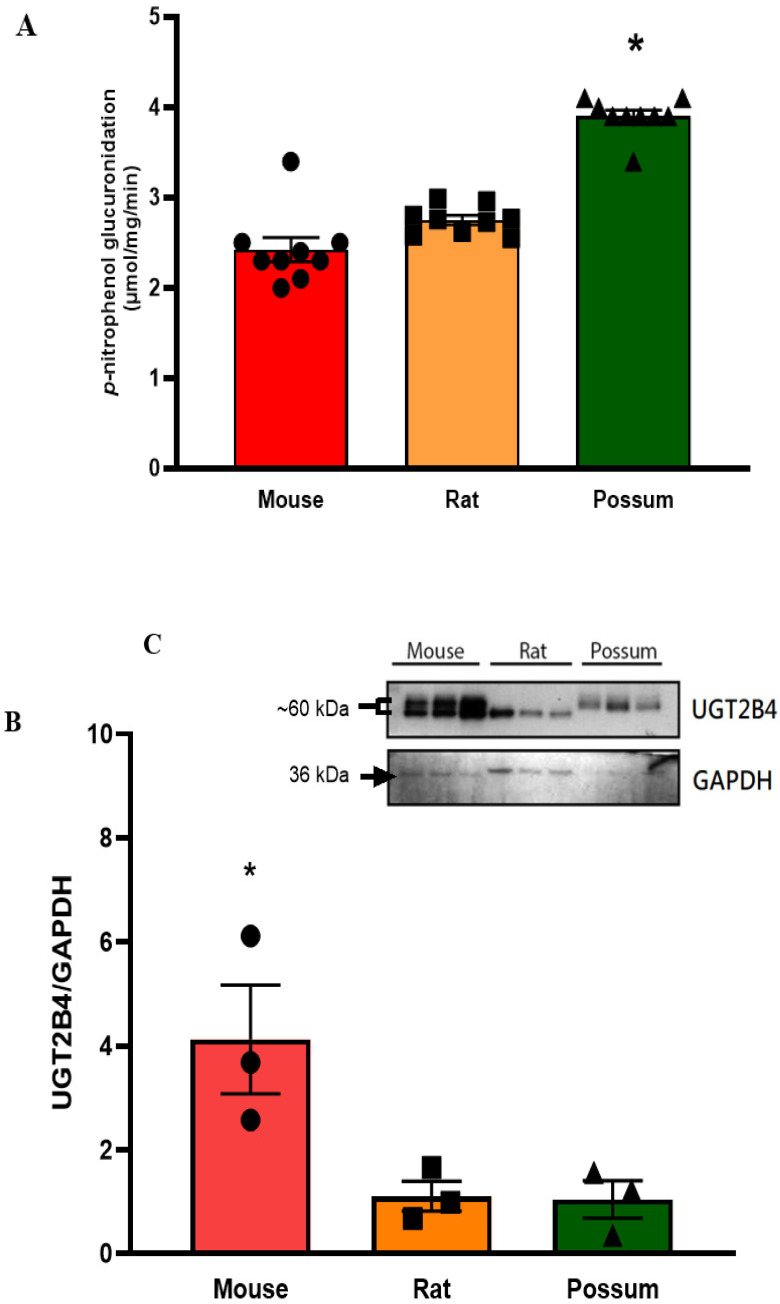
**Hepatic *p*-nitrophenol glucuronidation activity and protein levels in mouse, rat, and possum**: (**A**) Hepatic microsomes from each species were incubated with 0.1 M phosphate buffer and <0.5% DMSO, and *p*-nitrophenol glucuronidation activity was determined as an indicator of UGT catalytic activity. Values are expressed as µmol/mg/min. Bars represent the mean ± SEM from basal catalytic activities in each species. (**B**) Protein content for UGT2B4 in hepatic microsome from each species was determined by Western blotting (loaded protein in each well was 1 µg). Results from scanning densitometry of UGT2B4 Western blots normalised to the housekeeper protein GAPDH. The bars represent the mean ± SEM of the optical density from n = 3. (**C**) Representative image for UGT2B4 Western blot from left to right: mouse, rat, and possum, n = 3 * Significantly higher compared to the other 2 species, *p* < 0.05.

**Figure 5 biology-10-01326-f005:**
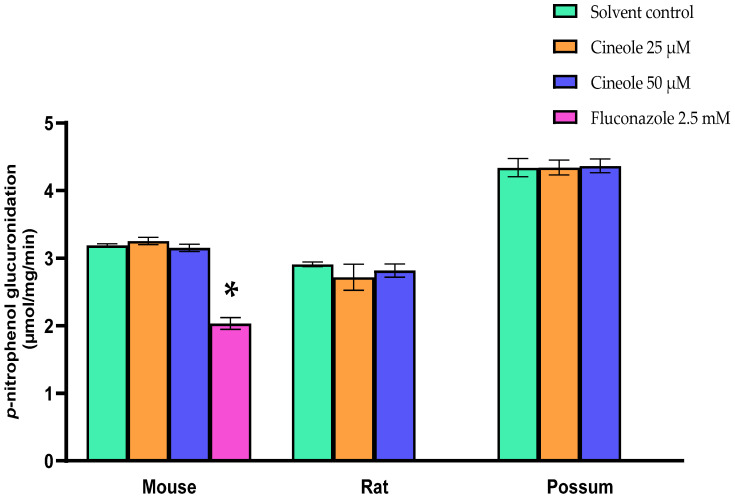
***P*-nitrophenol glucuronidation activity of cineole in mouse, rat, and possum**. Hepatic microsomes from each species were incubated with cineole (25 and 50 µM) and *p*-nitrophenol glucuronidation activity was determined as an indicator of UGT catalytic activity. Fluconazole was used as a positive control. Values are expressed as µmol/mg/min. Bars represent the mean ± SEM from n = 3 in duplicate. * Significantly decreased compared to solvent control in the mouse, *p* < 0.05.

**Table 1 biology-10-01326-t001:** Hepatic CYP3A catalytic activities following cineole and its derivatives in the mouse, rat, and possum.

Cineole and Its Derivatives	Species	Solvent Control	12.5 µM		25 µM	
		Activity *	Activity *	% of Control	Activity *	% of Control
**Cineole**	Mouse	0.68 ± 0.02	0.68 ± 0.02	**100.71** **± 2.93**	0.64 ± 0.02	**94.48** **± 2.69**
	Rat	0.64 ± 0.06	0.64 ± 0.05	**100.53** **± 2.31**	0.65 ± 0.05	**101.27** **± 1.62**
	Possum	0.40 ± 0.01	0.40 ± 0.01	**99.53** **± 2.67**	0.41 ± 0.01	**101.86** **± 0.29**
**4-pentylbicyclo[2.2.2]octane-1-carboxylic acid**	Mouse	0.63 ± 0.03	0.61 ± 0.04	**96.35** **± 1.91**	0.67 ± 0.01	**105.58** **± 2.97**
	Rat	0.53 ± 0.06	0.56 ± 0.05	**104.68** **± 3.30**	0.55 ± 0.06	**102.05** **± 2.42**
	Possum	0.36 ± 0.04	0.40 ± 0.05	**109.05** **± 3.58**	0.40 ± 0.02	**110.83** **± 7.26**
**5-norbornen-2-endo-3-exo-dicarboxylic acid**	Mouse	0.75 ± 0.03	0.73 ± 0.04	**96.48** **± 0.72**	0.77 ± 0.03	**101.88** **± 0.75**
	Rat	0.61 ± 0.10	0.65 ± 0.10	**107.52** **± 3.84**	0.64 ± 0.09	**105.30** **± 2.08**
	Possum	0.40 ± 0.02	0.39 ± 0.03	**98.90** **± 0.73**	0.38 ± 0.02	**95.23** **± 3.85**
**Cis-1,5-dimethylbicyclo[3.3.0]octane-3,7-dione**	Mouse	0.71 ± 0.04	0.71 ± 0.03	**100.94** **± 2.70**	0.70 ± 0.03	**99.64** **± 0.93**
	Rat	0.61 ± 0.06	0.59 ± 0.04	**98.31** **± 3.10**	0.60 ± 0.06	**99.32** **± 1.00**
	Possum	0.38 ± 0.02	0.41 ± 0.02	**105.83** **± 1.42**	0.40 ± 0.03	**102.79** **± 6.11**
**Trans-terpin**	Mouse	0.71 ± 0.04	0.71 ± 0.03	**100.94** **± 2.70**	0.70 ± 0.03	**99.64** **± 0.93**
	Rat	0.61 ± 0.06	0.59 ± 0.04	**98.31** **± 3.10**	0.60 ± 0.06	**99.32** **± 1.00**
	Possum	0.38 ± 0.02	0.41 ± 0.02	**105.83** **± 1.42**	0.40 ± 0.03	**102.79** **± 6.11**
**1,1-dimethylcyclohexane**	Mouse	0.74 ± 0.03	0.73 ± 0.03	**98.93** **± 0.94**	0.79 ± 0.04	**106.62** **± 1.13**
	Rat	0.61 ± 0.11	0.61 ± 0.07	**103.11** **± 7.68**	0.61 ± 0.10	**100.08** **± 1.66**
	Possum	0.39 ± 0.02	0.38 ± 0.01	**97.96** **± 1.28**	0.40 ± 0.02	**101.83** **± 0.61**
**7-oxabicyclo[2.2.1]-hept-5-ene-2-carboxylic acid**	Mouse	0.65 ± 0.01	0.67 ± 0.01	**103.33** **± 2.77**	0.66 ± 0.03	**103.01** **± 3.52**
	Rat	0.45 ± 0.05	0.48 ± 0.06	**107.45** **± 5.87**	0.51 ± 0.04	**113.77** **± 3.17**
	Possum	0.39 ± 0.02	0.41 ± 0.02	**106.38** **± 7.47**	0.40 ± 0.02	**103.08** **± 1.09**
**Dimethyl-trans-cyclohexane-1,4-dicarboxylate**	Mouse	0.72 ± 0.02	0.72 ± 0.02	**100.15** **± 1.31**	0.77 ± 0.05	**106.11** **± 4.88**
	Rat	0.63 ± 0.03	0.64 ± 0.02	**102.31** **± 2.42**	0.73 ± 0.06	**115.51** **± 5.35****^#^**
	Possum	0.38 ± 0.02	0.39 ± 0.02	**102.02** **± 4.76**	0.42 ± 0.03	**110.63** **± 6.00**
**Camphor**	Mouse	0.70 ± 0.02	0.68 ± 0.02	**97.10** **± 2.15**	0.68 ± 0.03	**96.29** **± 3.21**
	Rat	0.63 ± 0.05	0.63 ± 0.03	**100.68** **± 3.99**	0.59 ± 0.04	**95.04** **± 1.81**
	Possum	0.39 ± 0.02	0.40 ± 0.03	**102.38** **± 2.05**	0.41 ± 0.03	**104.15** **± 2.57**

* CYP3A catalytic activity expressed in nmol/mg/min. Each compound was tested in three biological replicates (n = 3 in duplicate). ^#^
*p* < 0.05 shows a significant difference in percent control activity compared to the control. However, none strongly reduced the catalytic activity compared to control.

**Table 2 biology-10-01326-t002:** Predicted CYP3A4 inhibition using cineole and its derivatives.

Compound Names	Predicted Values of CYP3A4 Inhibition
Ketoconazole (Positive control)	**1.0**
Cineole	**0**
4-pentylbicyclo[2.2.2]octane-1-carboxylic acid	**0**
5-norbornen-2-endo-3-exo-dicarboxylic acid	**0**
Cis-1,5-dimethylbicyclo[3.3.0]octane-3,7-dione	**0**
Trans-terpin	**0**
1,1-dimethylcyclohexane	**0**
7-oxabicyclo[2.2.1]hept-5-ene-2-carboxylic acid	**0**
Dimethyl-trans-cyclohexane-1,4-dicarboxylate	**0**
Camphor	**0**

‘0’ indicates no CYP3A4 inhibition.

**Table 3 biology-10-01326-t003:** Hepatic *p*-nitrophenol glucuronidation following cineole and its derivatives in the mouse, rat and possum.

Cineole and Its Derivatives	Species	Solvent Control	12.5 µM		25 µM	
		Activity *	Activity *	% of Control	Activity *	% of Control
**Cineole**	Mouse	2.00 ± 0.20	2.00 ± 0.16	**100.36** **± 1.99**	1.90 ± 0.16	**94.21** **± 1.51**
	Rat	2.73 ± 0.07	2.46 ± 0.12	**90.25 ± 6.49**	2.67 ± 0.21	**98.18** **± 9.75**
	Possum	4.11 ± 0.12	4.09 ± 0.18	**99.39** **± 1.71**	4.18 ± 0.13	**101.80** **± 1.12**
**4-pentylbicyclo[2.2.2]octane-1-carboxylic acid**	Mouse	2.45 ± 0.12	2.41 ± 0.06	**98.73** **± 3.51**	2.21 ± 0.14	**90.15** **± 2.18 ^#^**
	Rat	2.77 ± 0.03	2.88 ± 0.13	**103.95** **± 3.65**	2.82 ± 0.07	**101.89** **± 3.31**
	Possum	3.86 ± 0.23	3.80 ± 0.27	**98.44** **± 1.05**	3.96 ± 0.17	**102.71** **± 1.91**
**5-norbornen-2-endo-3-exo-dicarboxylic acid**	Mouse	2.09 ± 0.14	2.23 ± 0.09	**107.16** **± 4.97**	2.23 ± 0.10	**107.42** **± 3.63**
	Rat	2.96 ± 0.13	2.85 ± 0.16	**96.42** **± 1.23**	2.92 ± 0.19	**98.52** **± 2.14**
	Possum	3.87 ± 0.20	3.90 ± 0.23	**100.53** **± 2.09**	3.91 ± 0.20	**100.86** **± 1.90**
**Cis-1,5-dimethylbicyclo[3.3.0]octane-3,7-dione**	Mouse	2.50 ± 0.09	2.56 ± 0.11	**102.19** **± 2.82**	2.40 ± 0.15	**95.72** **± 2.42**
	Rat	2.62 ± 0.06	2.58 ± 0.03	**98.32** **± 3.38**	2.56 ± 0.02	**97.79** **± 2.87**
	Possum	4.00 ± 0.21	3.77 ± 0.28	**93.15** **± 4.58**	3.78 ± 0.28	**93.30** **± 4.13**
**Trans-terpin**	Mouse	2.39 ± 0.04	2.44 ± 0.06	**101.67** **± 0.74**	2.33 ± 0.07	**97.15** **± 1.80**
	Rat	2.76 ± 0.04	2.87 ± 0.22	**104.14** **± 9.44**	2.50 ± 0.10	**90.51** **± 3.63**
	Possum	3.92 ± 0.32	3.90 ± 0.34	**99.53** **± 0.84**	3.91 ± 0.34	**99.77** **± 0.60**
**1,1-dimethylcyclohexane**	Mouse	2.34 ± 0.07	2.48 ± 0.13	**105.97** **± 2.23**	2.19 ± 0.09	**93.67** **± 1.67**
	Rat	2.58 ± 0.11	2.59 ± 0.08	**100.53** **± 2.39**	2.58 ± 0.05	**100.29** **± 3.55**
	Possum	4.05 ± 0.20	4.08 ± 0.18	**100.68** **± 0.60**	4.03 ± 0.23	**99.32** **± 0.90**
**7-oxabicyclo[2.2.1]hept-5-ene-2-carboxylic acid**	Mouse	3.42 ± 0.06	3.46 ± 0.04	**101.00** **± 0.64**	3.26 ± 0.09	**95.53** **± 4.50**
	Rat	2.80 ± 0.07	2.87 ± 0.07	**102.50** **± 2.98**	2.75 ± 0.06	**98.20** **± 1.42**
	Possum	4.05 ± 0.07	4.02 ± 0.09	**99.13** **± 1.29**	3.99 ± 0.09	**98.49** **± 0.83**
**Dimethyl-trans-cyclohexane-1,4-dicarboxylate**	Mouse	2.28 ± 0.06	2.43 ± 0.06	**107.00** **± 0.42**	2.73 ± 0.12	**120.23** **± 8.23 ^#^**
	Rat	2.99 ± 0.16	2.71 ± 0.13	**90.77** **± 2.49**	2.83 ± 0.20	**94.83** **± 5.80**
	Possum	3.86 ± 0.21	3.96 ± 0.20	**102.94** **± 4.63**	3.92 ± 0.19	**101.67** **± 0.92**
**Camphor**	Mouse	2.31 ± 0.08	2.42 ± 0.08	**104.80** **± 1.79**	2.31 ± 0.09	**100.10** **± 2.45**
	Rat	2.55 ± 0.06	2.62 ± 0.09	**102.59** **± 1.43**	2.72 ± 0.24	**106.26** **± 6.99**
	Possum	3.93 ± 0.38	4.01 ± 0.33	**102.42** **± 1.78**	3.97 ± 0.33	**101.32** **± 1.34**

*  *p*-nitrophenol glucuronidation activity expressed in µmol/mg/min (n = 3 in duplicate). Each compound was tested in three biological replicates (n = 3 in duplicate). ^#^
*p* < 0.05 shows a significant difference in percent control activity compared to the control. However, none strongly reduced the catalytic activity compared to control.

## Data Availability

The data presented in this study are reported in the results and Appendix A.

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
