# Peer review of "In Vitro Hepatic Assessment of Cineole and Its Derivatives in Common Brushtail Possums (Trichosurus vulpecula) and Rodents"

_biology, 2021, doi:10.3390/biology10121326_

Round 1

Reviewer 1 Report

I appreciate the excellent writing. I have reviewed too many papers that have been written poorly. Well done on this one. The study is well-designed and experiments, as described, well executed based on the clean results. I wish you well as you explore how to leverage the variances you observed between the possum and rodent enyzme activities and expression. I inetns to have my biochemistry students read this paper after it is published as a case study for good design.

Author Response

Thank you so much for this wonderful comment!

Attached is the revised version of the manuscript. Please note the text in blue shows new information added to the original manuscript. 

Reviewer 2 Report

Comments:

In this manuscript, the authors described “In vitro hepatic assessment of cineole and its derivatives in common brushtail possums (Trichosurus vulpecula) and rodents.”. This study shows the effect of cineole and its derivatives on hepatic CYP3A and p-nitrophenol catalytic activity in mice. However, there are a few points that need to be clarified.

Comment

  1. The author should strengthen silico modelling and UDP-glucuronosyltransferas of the narrative in this discussion.

  1. The other scientific paper also investigated (1-3) the metabolism of cineole (1,8-Cineole). The author shall be discussion.

Reference:

  1. Miyazawa M, Shindo M, Shimada T. Oxidation of 1,8-cineole, the monoterpene cyclic ether originated from eucalyptus polybractea, by cytochrome P450 3A enzymes in rat and human liver microsomes. Drug Metab Dispos. 2001 Feb;29(2):200-5.
  2. Pass GJ, McLean S. Inhibition of the microsomal metabolism of 1,8-cineole in the common brushtail possum (Trichosurus vulpecula) by terpenes and other chemicals. Xenobiotica. 2002 Dec;32(12):1109-26.
  3. Zhang W, Lim LY. Effects of spice constituents on P-glycoprotein-mediated transport and CYP3A4-mediated metabolism in vitro. Drug Metab Dispos. 2008 Jul;36(7):1283-90.

Author Response

The author should strengthen in silico modelling and UDP-glucuronosyltransferase of the narrative in this discussion.

Response- We have taken reviewer’s comments in full consideration. I have added further details in the discussion section as follows:

  1. In silico study (line 341-346)

Moreover, in silico study using the DL-CYP Prediction Server has further supported that cineole and derivatives have no inhibitory action on the CYP3A4 isoform. The predicted values for cineole and its derivatives on CYP3A4 inhibition were 0; indicating that these compounds would reduce catalytic activity at higher concentrations (> 57 µM) and therefore were considered to be noninhibitors.

Similarly, additional information was also added to the methodology section (2.5. Prediction of CYP Inhibition in silico) (line 161-166). The AC50 value (concentration that gives 50% activation of the enzyme) was used as one of the main criteria to group compounds either as an inhibitor or noninhibitor [17]. Compounds with AC50 value ≤ 10 µM were grouped as inhibitors while compounds with AC50 value > 57 µM were considered noninhibitors.

  1. UDP-glucuronosyltransferase enzyme (line 388-391)

The present study showed that the basal p-nitrophenol glucuronidation activities in the possum were ~2-fold higher than those in mouse and rat hepatic microsomes; indicating a greater capacity to metabolise phenolic compounds. Higher p-nitrophenol glucuronidation in possum liver microsomes may also explain why marsupial folivores such as the koala can rapidly excrete 2-3 g of glucuronic acids daily [33].

  1. The other scientific paper also investigated (1-3) the metabolism of cineole (1,8-Cineole). The author shall be discussion.
  2. Miyazawa M, Shindo M, Shimada T. Oxidation of 1,8-cineole, the monoterpene cyclic ether originated from eucalyptus polybractea, by cytochrome P450 3A enzymes in rat and human liver microsomes. Drug Metab Dispos. 2001 Feb;29(2):200-5.

Response- This reference was added in the discussion section (CYP3A catalytic activity) (line 358-361): Similarly, Miyazawa, et al. [29] also suggested that cineole is one of the effective substrates for CYP3A enzymes in the rat and human liver microsomes.

  1. Pass GJ, McLean S. Inhibition of the microsomal metabolism of 1,8-cineole in the common brushtail possum (Trichosurus vulpecula) by terpenes and other chemicals. Xenobiotica. 2002 Dec;32(12):1109-26.

Response- This reference was already in the manuscript.

  • Zhang W, Lim LY. Effects of spice constituents on P-glycoprotein-mediated transport and CYP3A4-mediated metabolism in vitro. Drug Metab Dispos. 2008 Jul;36(7):1283-90.

Response-This reference was added in the discussion section (CYP3A Catalytic activity) (line 340-341): However, greater cineole concentration (>1 mM) may potentially reduce the catalytic activity [26].